# AxisGuide: Grounding Robot Action Coordinate System in RGB Observations for Robust Visuomotor Manipulation

*Abstract*— **Visuomotor manipulation policies trained via large-scale behavior cloning have achieved strong semantic scene understanding, yet often fail to reliably execute correct low-level actions under distribution shifts. For example, even in a simple pick-up task with identical scene layouts, camera viewpoints, and illumination, performance can degrade substantially when the object is placed at unseen locations. We argue that this gap arises from insufficient action understanding, namely the inability to interpret the robot's base-frame action coordinate system in image space. To address this issue, we introduce AxisGuide, a lightweight guidance method that bridges semantic scene understanding and action-coordinate interpretation. Using camera parameters and end-effector poses, AxisGuide renders the robot base-frame axes in each camera view and augments RGB observations with a small set of cue channels that explicitly visualize the meaning of $+x/+y/+z$ motions in image space. Extensive evaluations in both the LIBERO simulation and real-world environments demonstrate that AxisGuide yields substantial performance gains and improved generalization, highlighting the effectiveness of explicit action-coordinate cues for learning reliable and transferable generalist visuomotor policies. Our demo video is available at [this link].**

## I. INTRODUCTION

Recent visuomotor manipulation policies [1], [2], [3] have achieved impressive performance by scaling behavior cloning with large collections of demonstrations. Building on this foundation, Vision-Language-Action (VLA) models extend visuomotor learning by leveraging vision-language pretraining to improve semantic understanding, enabling broader generalization across tasks [4], [5], [6], [7], [8]. Nevertheless, we still observe a persistent gap between understanding and executing: even in situations where a model appears to semantically "understand" scenes, it often fails to reliably translate that understanding into correct low-level actions [9]. In other words, the policy may capture *what* to do at a high level, but it often struggles with *how* to execute it robustly. This gap naturally leads to a fundamental question: **Do visuomotor policies truly understand actions?**

In manipulation, "understanding actions" is closely tied to understanding the *action coordinate system*. Most modern visuomotor policies perceive the world through RGB images and infer actions from pixels. In most setups [10], [6], [4], actions are defined as relative end-effector (EEF) poses with respect to the robot base frame, typically parameterized as 6D $(x, y, z, \text{roll}, \text{pitch}, \text{yaw})$ plus a gripper command. Because a visuomotor policy infers actions from pixels and a 6D action is composed of independent unit dimensions, successful task execution requires understanding what each action dimension means in image and how to combine them to generate appropriate behavior in novel situations. Shortly,

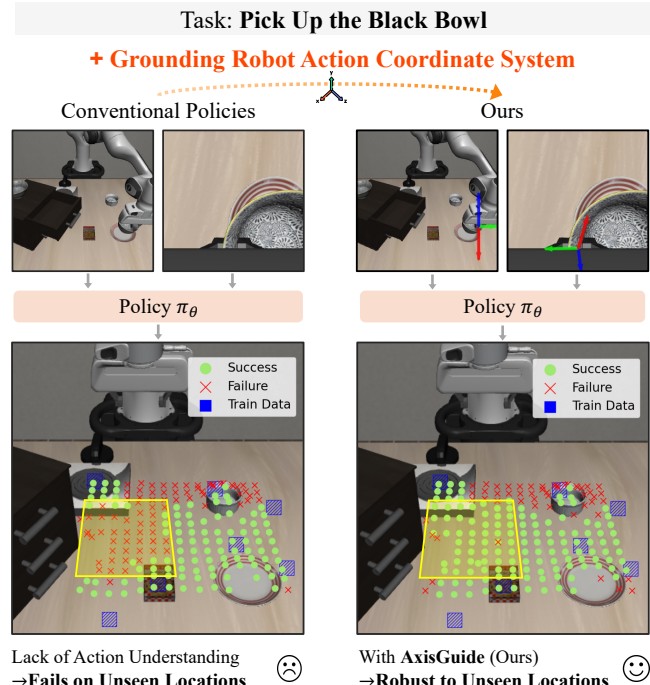

Fig. 1: **AxisGuide: Grounding Robot Action Coordinate System for Robust Manipulation.** Conventional visuomotor policies (left) struggle to generalize beyond training data (blue squares), often failing at unseen locations (yellow box). In contrast, AxisGuide (right) enables robust task execution across a wide range of unseen spatial configurations. By explicitly associating the action space with image observations through grounding the robot action coordinate system in image space, AxisGuide bridges the gap between semantic scene understanding and action coordinate interpretation.

a policy must recognize how the action coordinate system manifests in the RGB observation to behave robustly. For example, the model must know which direction in the RGB image corresponds to $+x$ in the action space and which rotation corresponds to yaw in the given camera view, and use this knowledge to execute the intended actions.

However, it is inherently difficult for policies to acquire such correspondence between RGB image and action space under the standard observation-to-action setting, because RGB images and robot states rarely provide an explicit direct reference that anchors the action coordinate system. Although the robot base serves as a reference, it is often outside the camera view or occluded in practice [11], [12], [13]. Therefore, behavior cloning often degenerates into memorizing correspondences between images and numeric

action vectors, which leads to poor generalization.

To illustrate this degeneration, we train a VLA model [8] on a simple pick-up task with randomized object placements, using both a wrist-mounted and a front camera in a setup where the robot base frame is clearly visible, while keeping the instruction and manipulation objective fixed and varying only the object position. We then evaluate whether the policy can pick up the same object at unseen locations (Fig. 1, left). This controlled setting minimizes confounding effects from task semantics and enables a focused evaluation of action understanding, since successful generalization requires the policy to compose new relative end-effector displacements in response to target shifts. Despite consistent object appearance, the model often fails to adapt its motion to shifted targets. Similar trends are reported in concurrent work [14], where performance degrades under object position changes. These results suggest that without grounding the action coordinate system in image space, policies struggle to generalize beyond the training distribution.

This motivates providing the policy with an explicit association between the action spaces and image observations. Therefore, we propose **AxisGuide**, a lightweight guidance approach that explicitly bridges the action space and its meaning in RGB observations. Using camera parameters, we render the axes of the robot base frame in each camera view and augment each RGB observation with a small set of additional channels that encode these visual cues, as illustrated in Fig. 2. Concretely, for each view we visualize "what $+x/+y/+z$ motion means in the image" by projecting three unit base-frame translations onto the image plane. We take unit translation vectors $(\Delta x, \Delta y, \Delta z) \in \{(1,0,0),(0,1,0),(0,0,1)\}$ from the action space and render their corresponding 2D direction vectors on EEF pose in image space. These vectors are simply computed from camera parameters and the end-effector pose without requiring depth, and concatenated with the original RGB image in a channel-wise manner as input to the policy. As illustrated in Fig. 1 (right), by making the meaning of the action space explicit in pixels, AxisGuide enables policies to adapt their motion more reliably to novel situations. Quantitatively, AxisGuide improves success rates at unseen object locations by up to 20%p across real-world and simulation [15] experiments. We observe consistent gains in both multi-view and single-view settings, suggesting that explicit action-coordinate grounding leads to more reliable, goal-directed execution.

In summary, our contributions are as follows:

- We identify a previously underexplored gap between semantic understanding and action execution in current visuomotor policies, and define it as the challenge of understanding action coordinate systems in image space.
- We propose AxisGuide, a simple yet effective solution that injects explicit references for the action coordinate systems in RGB images into visual observations.
- We demonstrate consistent gains in performance and generalization across both simulated and real-world

manipulation tasks under diverse camera configurations.

## II. METHODOLOGY

### A. Method Overview

A key challenge in learning image-to-action policies is that actions are defined in a robot-centric coordinate system (e.g., base frame), while observations (RGB images and proprioception) provide no explicit reference for interpreting each action dimension $(x, y, z)$ in the current view. As a result, the policy must implicitly infer the mapping between 6D robot actions and their visual consequences in image space purely from data, which can lead to poor generalizability under distribution shifts. AxisGuide addresses this mismatch by making the action coordinate system visually explicit (Fig. 2). We project the robot base-frame axes into image space and provide them as additional visual cues, allowing the policy to directly interpret how actions such as "move along $+x$" appear in the current view. Concretely, at each timestep we render three direction vectors from the current end-effector location, corresponding to unit translations along the $x$, $y$, and $z$ axes. These are encoded as a $3 \times H \times W$ cue image $A_t$, where each axis is represented in the red, green, and blue channels, respectively, and concatenated channel-wise with the RGB input. Details of the rendering process are provided in the next section.

### B. Computing Action Coordinate Cue Image.

Let $\mathbf{p}_w \in \mathbb{R}^3$ be the current gripper position in the robot-base/world frame, and let $\Pi(\cdot)$ be the camera projection defined by the intrinsics/extrinsics following Eq. (4) and Eq. (5) in Appendix. We first obtain the pixel location of the gripper by projecting $\mathbf{p}_w$ onto the image plane, $\mathbf{o} = \Pi(\mathbf{p}_w) \in \mathbb{R}^2$. To compute image-space direction vectors corresponding to unit translations along each base axis, we consider three canonical unit moves $\Delta\mathbf{p}_w^{(x)} = \varepsilon\mathbf{e}_x$, $\Delta\mathbf{p}_w^{(y)} = \varepsilon\mathbf{e}_y$, $\Delta\mathbf{p}_w^{(z)} = \varepsilon\mathbf{e}_z$, where $\mathbf{e}_x, \mathbf{e}_y, \mathbf{e}_z$ are the robot base-frame basis vectors and $\varepsilon > 0$ is a small step size. We project the translated points onto the image plane and compute the corresponding image space displacements as

$$\Delta\mathbf{u}^{(k)} = \Pi\Big(\mathbf{p}_w + \Delta\mathbf{p}_w^{(k)}\Big) - \Pi(\mathbf{p}_w), k \in \{x, y, z\}. \quad (1)$$

Then we normalize them to obtain unit image-space direction vectors $\hat{\mathbf{d}}^{(k)}$. Finally, we render three arrows starting at $\mathbf{o}$ and pointing along $\hat{\mathbf{d}}^{(x)}, \hat{\mathbf{d}}^{(y)}, \hat{\mathbf{d}}^{(z)}$, using red/green/blue channels of an RGB image for $x/y/z$, respectively. We anchor these arrows at $\mathbf{o}$ (the projected end-effector pixel) to emphasize the gripper's current interaction point and to provide an end-effector-centric directional reference. Rendering the $x/y/z$ directions at the moving EEF explicitly visualizes where the robot will move from its current state in the image, making the action semantics directly observable. The rendered result forms an action coordinate cue image $A_t \in \mathbb{R}^{3 \times H \times W}$.

### C. Policy Learning With Action Coordinate Cue Image.

We inject the action-coordinate cue image via channel-wise concatenation rather than overlaying them onto the RGB image observation. Overlays can occlude small but

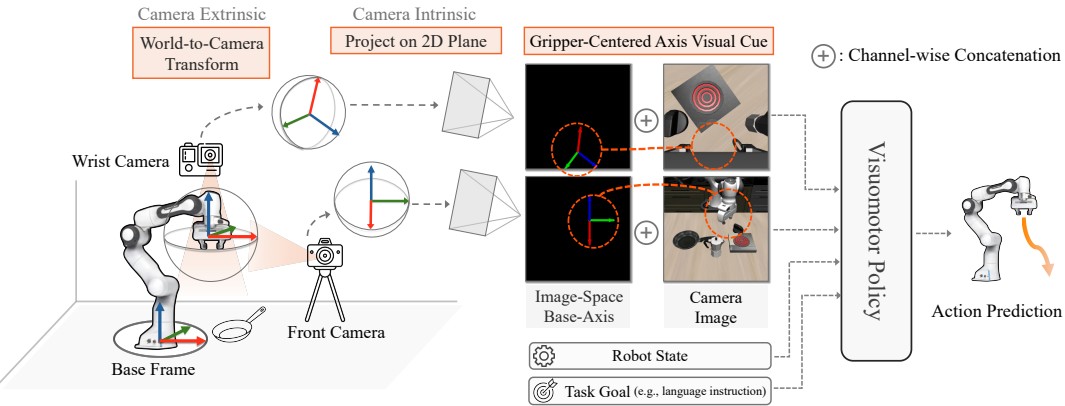

Fig. 2: **An overview of AxisGuide.** Using camera intrinsics and extrinsics, AxisGuide projects the robot base-frame $x$, $y$, and $z$ axes onto the 2D image plane, centered at the gripper, and renders them as additional channels alongside RGB images from all cameras. This explicit visualization enables the policy to better understand the correspondence between visual observations and robot base-frame actions.

critical visual details and alter the raw appearance distribution, which may interfere with perception. Channel-wise concatenation preserves the original RGB content while providing a dedicated cue channel for the policy to exploit. Following the common practice of injecting additional visual tokens/channels by early fusion [16], [17], AxisGuide also augments each view by concatenating the RGB image with the action-coordinate cue image channel-wisely:

$$\tilde{I}_t^{(v)} = [I_t^{(v)}; A_t^{(v)}] \in \mathbb{R}^{6 \times H \times W}, \tilde{o}_t = \left( \{\tilde{I}_t^{(v)}\}_{v=1}^V, q_t \right), \quad (2)$$

where $v$ indexes camera views, $I_t$ denotes the image input (single-view or multi-view), and $q_t$ denotes the proprioceptive state (e.g., joint positions). To process $\tilde{I}_t^{(v)}$, we simply increase the input channel dimension of the vision backbone [18], [19], [20]'s first convolution layer from 3 to 6 (per view), while keeping the rest of the backbone and the policy head unchanged. We then train the policy with the standard behavior cloning objective [21]:

$$\max_\theta \ \mathbb{E}_{(a_{t:t+H}, \tilde{o}_t, c) \sim \mathcal{D}} \left[ \log \pi_\theta(a_{t:t+H} \mid \tilde{o}_t, c) \right]. \quad (3)$$

The conditioning variable $c$ is optional and depends on the policy.

## III. EXPERIMENT

In this section, we present multiple experiments to evaluate the effectiveness of AxisGuide with detailed analysis addressing the following key research questions:

Q1. (Section IV-A) Does AxisGuide help the policy understand action coordinate systems?

Q2. (Section IV-B) Does AxisGuide lead to better control and higher task success rate?

Experimental setups, including simulation and real-world environments, tasks, and policy and baseline implementations, are provided in the Appendix (Sec. VI-A and Sec. VI-B).

### A. Does AxisGuide Help the Policy Understand Action Coordinate Systems?

In this section, we evaluate whether AxisGuide encourages policies to learn action semantics from visual observations.

To test whether a policy correctly interprets the meaning of actions, we construct a dataset in which the same object is placed at multiple locations. To only evaluate action-coordinate understanding for evaluation while minimizing confounding factors such as long-horizon planning and complex object interactions, we adopt **Pick Up** as a simple manipulation task. If a policy truly learns the task objective and the semantics of its actions, it should successfully pick up the object regardless of its location. We collect demonstrations across diverse object positions under a multi-view configuration using both front and wrist-mounted cameras.

In simulation, we use the official LIBERO-Spatial dataset, which involves picking up a black bowl from varying locations, and evaluate performance on the **Pick Up (Bowl)** task. We collect 400 demonstrations from the LIBERO-Spatial suite and report performance as the average success rate over 210 evaluation rollouts on unseen locations, as illustrated in Fig. 1. In the real world, using the corresponding setup shown in Fig. 3 (a), we evaluate DP on the **Pick Up (Pear)** task to verify that the observed effects are not model-specific. We collect 120 demonstrations and evaluate performance over 84 rollouts on unseen locations.

The quantitative results in Fig. 3 (b) suggest that AxisGuide enables policies to better reach objects not only at seen positions but also at novel object locations. Specifically, AxisGuide improves the simulation success rate from 52.38% to 65.71% (+13.33%p) and the real-world success rate from 30.12% to 50.00% (+19.88%p). Qualitative results in Fig. 1 and Fig. 3 (a) further show that baseline models generalize poorly beyond the training data, whereas models trained with AxisGuide reliably reach unseen object positions between clusters in both simulation and real-world settings.

To better understand these results, we further visualize representative rollout trajectories from the evaluation set (Fig. 8 in the Appendix). In real-world and simulation, SmolVLA exhibits weaker robustness to object location shifts, often drifting toward a training-like configuration even when the bowl is relocated, whereas AxisGuide produces a trajectory that redirects toward the shifted target and reaches it more

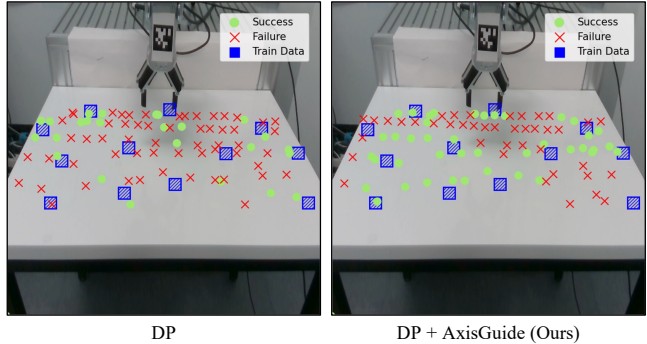

(a) Spatial Generalization Across Object Positions in Real World

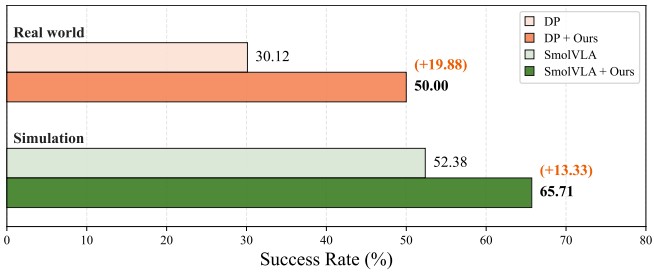

(b) Comparison of Success Rates in Real-World and Simulation

Fig. 3: **Generalization to Novel Object Positions.** (a) shows that the baseline DP (left) generalizes poorly, with success largely confined to regions near training data, whereas DP with AxisGuide (right) reliably reaches unseen object positions between clusters, which is consistent with the simulation results in Fig. 1. We report corresponding success rates in (b), where AxisGuide delivers substantial gains in both real-world and simulation settings.

TABLE I: **Quantitative Comparison of Single-View Visuomotor Policies in the LIBERO Simulation.** We evaluate average success rates (%) over 75 rollouts (25 rollouts × 3 seeds). See the Sec III-B for further discussion.

| Method | Pick & Place (Bowl) | Drawer | Stove |
|---|---|---|---|
| DP [1] | 69.33 | 86.67 | 92.00 |
| DP + KYC [17] | 73.33 | 89.33 | 96.00 |
| DP + Ours | **82.67** (13.34↑) | **93.33** (6.66↑) | **100.0** (8.00↑) |

precisely. Overall, these results indicate that AxisGuide encourages policies to produce appropriate action adjustments by grounding action coordinates in pixels, leading to more robust behavior under distribution shift.

### B. Does AxisGuide Lead to Better Control and Higher Task Success Rate?

In this section, we evaluate whether additional information from AxisGuide's action coordinate cues actually help the policy to successfully execute desired tasks. Since many manipulation systems are limited to a single external camera, we test whether AxisGuide is effective in the front-camera-only setting. As a point of comparison for anchoring action semantics in image space, we adopt **Know Your Camera**

TABLE II: **Quantitative Comparison of Visuomotor Policies in Real-World.** We evaluate average success rates (%) over 30 rollouts. The single-view setup uses a fixed front camera, while the multi-view setup includes an additional wrist-mounted camera. More details about tasks are provided in Appendix (Sec VI-A)

| View type | Method | Pick & Place (Grape) | Flip Pot | Close Pot |
|---|---|---|---|---|
| Multi-view | DP | 93.39 | 73.33 | 36.66 |
| | DP + Ours | **96.66** (3.27↑) | **93.33** (20.00↑) | **53.33** (16.67↑) |
| Single-view | DP | 83.33 | 36.65 | 33.33 |
| | DP + KYC | 86.67 | 23.33 | 40.00 |
| | DP + Ours | **93.33** (10.00↑) | **50.00** (13.35↑) | **40.00** (6.67↑) |

**(KYC)** [17] as an additional baseline in the single-view setup, which explicitly conditions the policy on camera information.

We evaluate AxisGuide in both simulation and the real world using a single-view Diffusion Policy (DP). In simulation, we consider three LIBERO tasks spanning diverse action types and control skills: (1) **Pick & Place (Bowl)**: picking up the black bowl between the plate and the ramekin and placing it on the plate, (2) **Drawer**: opening the middle drawer of the cabinet, and (3) **Stove**: turning on the stove. In the real world, we follow the task setup in Appendix (Sec. VI-A). For each task, we train DP and report the average success rate over 75 evaluation rollouts in simulation (25 rollouts × 3 seeds) and 30 rollouts in the real world.

As shown in Table I, AxisGuide yields consistent improvements across all simulation tasks, with the largest gain on **Pick & Place (Bowl)**, where success depends on accurately perceiving small position shifts and executing precise corrections. Compared to the RGB-only baseline, AxisGuide improves success rate by 13.34%p and also outperforms KYC [17] by 9.34%p. Also, Table II (Single-view) shows the same trend in the real world, with particularly large gains on **Pick & Place (Grape)** and **Flip Pot**.

Overall, these results suggest that AxisGuide's explicit coordinate reference improves reliable task execution across diverse manipulation behaviors, boosting success in both simulation and real-world settings.

### IV. CONCLUSION

We study why visuomotor policies can exhibit strong semantic understanding yet fail to execute correct actions under distribution shifts, and show that performance drops significantly at unseen object locations even under identical conditions. We attribute this to insufficient action understanding. To address this, we propose **AxisGuide**, which explicitly visualizes the robot's base-frame action coordinate system in each view. By rendering unit motion directions as additional input channels, AxisGuide links visual observations to 3D action semantics without requiring depth, extra supervision, or architectural changes. Across real-world and simulation experiments, it consistently improves task success and robustness under distribution shifts.

## V. PRELIMINARIES

Given the camera extrinsics $\mathbf{T}_{c\leftarrow w} \in \mathbb{R}^{4\times 4}$ (world-to-camera), we represent the correspondence between a 3D point in the world frame $\mathbf{p}_w \in \mathbb{R}^3$ and the one in the camera frame $\mathbf{p}_c \in \mathbb{R}^3$ as

$$\begin{bmatrix} \mathbf{p}_c \\ 1 \end{bmatrix} = \mathbf{T}_{c\leftarrow w} \begin{bmatrix} \mathbf{p}_w \\ 1 \end{bmatrix}. \tag{4}$$

We assume a pinhole camera model [22] with intrinsics $\mathbf{K}$, under which a 3D point $\mathbf{p}_c = (X, Y, Z)$ with $Z > 0$ is projected onto the image plane as $\mathbf{p}_i = (u, v)$, where

$$\mathbf{K} = \begin{bmatrix} f_x & 0 & c_x \\ 0 & f_y & c_y \\ 0 & 0 & 1 \end{bmatrix}, \qquad \begin{aligned} u &= f_x \frac{X}{Z} + c_x, \\ v &= f_y \frac{Y}{Z} + c_y. \end{aligned} \tag{5}$$

## VI. ADDITIONAL EXPERIMENT

### A. Real World Setup Details

Fig. 5 shows the UR5e robot setup used in our real-world experiments. Each real-world task is evaluated to test the policy's ability to handle fine-grained control. All tasks use a 10Hz control frequency with RGB observations resized to $256 \times 256$. To verify that AxisGuide transfers to real-world deployment, we design a set of tabletop manipulation tasks that require precise action coordinate grounding and contact-rich control. Below, we describe the task setups in detail, including data collection protocols, randomization ranges, episode lengths, and success criteria.

1) **Pick & Place (Grape)**
   - **Task:** Picking up a grape from a random initial position on the table and placing it onto a target plate.
   - **Dataset:** 50 teleoperated demonstrations. To introduce spatial variety, the *plate* is sampled from a predefined planar region with jittering in the left–right and front–back directions. The *grape* is placed within a specific workspace, varying primarily along the front–back axis relative to the robot.
   - **Evaluation:** 30 trials are conducted. The initial positions of both the grape and the plate are randomized within the same spatial boundaries and perturbation ranges used during the data collection phase.
   - **Evaluation:** 30 trials are conducted. The initial positions of both the grape and the plate are randomized within the same spatial boundaries and perturbation ranges used during the data collection phase.
   - **Episode Length:** 140 timesteps (14 seconds).
   - **Success Criterion:** The task is successful if the grape remains stably on the plate in the final equilibrium state. We define success as the **Center of Mass (CoM)** of the grape staying within the

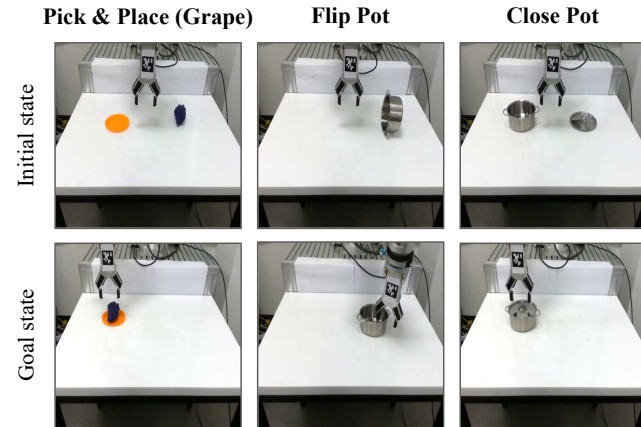

Fig. 4: **Real-world Manipulation Tasks for Evaluation.** We show initial states (top) and goal states (bottom) for **Pick & Place (Grape)**, **Flip Pot**, and **Close Pot**, which require different combinations of translational and rotational actions.

plate's boundary. If the CoM lies outside, the resulting instability causes the plate to tilt or the grape to fall, which is recorded as a failure.

2) **Flip Pot**
   - **Task:** Grasping the rim of a pot lying on its side and flipping it to an upright position on the table.
   - **Dataset:** 50 teleoperated demonstrations. The *pot* is initially placed within a designated workspace, where its position is perturbed with small translations (left–right and front–back) to encourage robust grasping under slight pose shifts.
   - **Evaluation:** 30 trials using the same predefined region and randomization protocol as the training set.
   - **Episode Length:** 220 timesteps (22 seconds).
   - **Success Criterion:** The pot must stand upright on its base without tipping over. This task evaluates the policy's understanding of rotational control and object pose.

3) **Close Pot**
   - **Task:** Picking up the pot lid and precisely aligning it with the pot opening to close it.
   - **Dataset:** 50 teleoperated demonstrations. The *pot* is placed within a predefined region and its position is slightly jittered along the left–right and front–back axes. The *lid* is placed in a fixed, reachable starting area.
   - **Evaluation:** 30 trials where the pot's initial position is randomized within the identical workspace used for the demonstrations.
   - **Episode Length:** 160 timesteps (16 seconds).
   - **Success Criterion:** The task is considered successful if both of the following conditions are met: (1) the gripper accurately grasps the handle of the pot lid without, and (2) the lid is seated on the pot covering at least 70% of the opening area. This evaluates fine-grained gripper-object

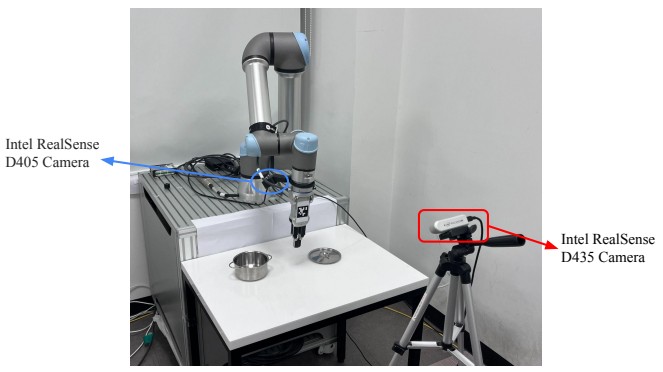

Fig. 5: **Real-world robot setup.** Our experiments are conducted on a UR5e arm with an RG2 gripper. The observation is captured with a fixed front camera and a wrist-mounted camera, providing complementary viewpoints of the scene for both single-view and multi-view evaluations. All RGB images are resized to $256 \times 256$ and the robot is controlled at $10\,\mathrm{Hz}$.

alignment and precise placement in contact-rich settings.

4) **Pick Up (Pear)**
   - **Task:** Reaching and picking up a pear placed at novel, unseen locations to test coordinate system understanding.
   - **Dataset:** 120 demonstrations collected in specific spatial clusters. We exclude areas within 5cm of the robot base or where the object is occluded.
   - **Evaluation:** 84 trials focused on unseen locations between or outside the training clusters, as shown in Fig. 3(a).
   - **Episode Length:** 50 timesteps (5 seconds).
   - **Success Criterion:** Successful grasp and lift of the pear at least 0.71m above the table surface.

### B. LIBERO Simulation Benchmark

We evaluate our method in the LIBERO [15] simulation benchmark, which provides language-conditioned robot manipulation tasks. Each task is specified by an initial-state distribution and a sparse goal predicate, and an episode terminates once all goal predicates are satisfied. LIBERO is designed to systematically study knowledge transfer in lifelong learning for decision-making by disentangling shifts in declarative knowledge (objects and spatial relationships) versus procedural knowledge (motions and behaviors). LIBERO comprises four task suites that capture different types of distribution shifts; we summarize them below and refer to Fig. 6 for qualitative examples of representative tasks from each suite.

**LIBERO-Spatial (10 tasks).** LIBERO-Spatial is designed to isolate transfer of *spatial knowledge*. Across the 10 tasks, the manipulation objective is kept largely consistent (e.g., placing a bowl onto a plate), while the key variation is *where the target object is located* and how it is situated relative to other objects in the scene. Concretely, the suite

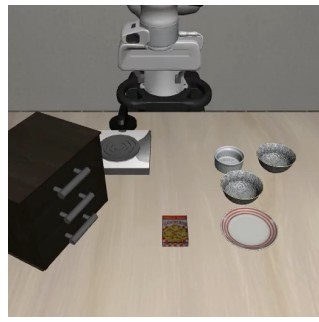 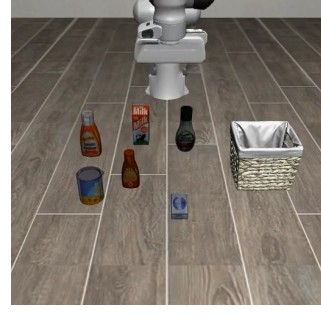

LIBERO-Spatial      LIBERO-Object

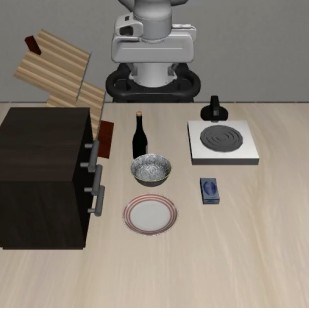 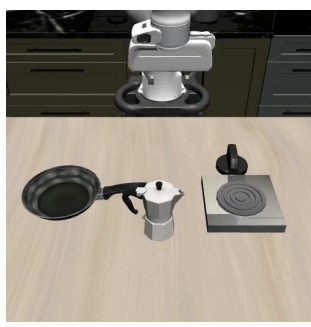

LIBERO-Goal      LIBERO-Long

Fig. 6: **LIBERO simulation benchmark [15].** We evaluate Diffusion Policy [1] and SmolVLA [8] on LIBERO task suites, and additionally augment each policy with **AxisGuide** cues to study action coordinate grounding. We construct the *object novel position* generalization benchmark using the LIBERO Spatial suite.

often includes visually identical instances (e.g., two similar bowls) whose only difference is their spatial configuration; thus, solving each new task primarily requires learning and retaining new *spatial relationships* rather than new object semantics or entirely new skills.

**LIBERO-Object (10 tasks).** LIBERO-Object targets transfer of *object knowledge*. Here, the overall environment structure and interaction pattern are kept similar, but each task introduces a *different target object* to be manipulated (e.g., "pick up *X* and place it *Y*" with varying *X*). As a result, the agent must continually acquire and retain new *visual and language grounding* for novel objects while reusing largely similar low-level manipulation behaviors.

**LIBERO-Goal (10 tasks).** LIBERO-Goal focuses on transfer of *procedural knowledge*. In this suite, tasks share the same set of objects and maintain fixed spatial relationships, while the primary change is the *goal specification* described by the language instruction and the corresponding goal predicates. This setting reduces the need for learning new objects or new layouts and instead emphasizes learning new *behaviors and action sequences* (i.e., procedural skills) to satisfy different goals.

**LIBERO-Long.** To evaluate long-horizon compositional manipulation, we additionally consider the long-horizon subset of LIBERO, referred to as LIBERO-Long, which consists of tasks that require *multi-stage* behavior and longer tem-

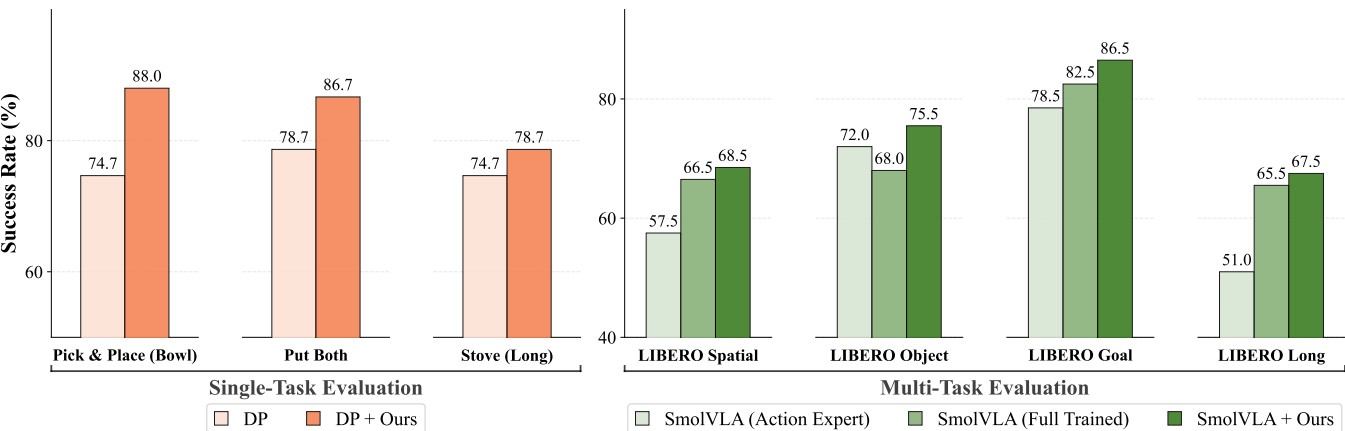

Fig. 7: **Quantitative Results in the Multi-View Simulation Setup (LIBERO).** We compare success rates of AxisGuide with baseline methods in single-task (left) and multi-task (right) settings using wrist and front cameras. Unlike the standard SmolVLA training pipeline [8], we train the full model including the image backbone to support additional coordinate cue channels. For fair comparison, we report both the action-expert-only and fully trained variants of SmolVLA. AxisGuide consistently outperforms Diffusion Policy (DP) [1] and SmolVLA across all evaluated tasks.

poral credit assignment (e.g., sequences of interactions such as opening/closing, inserting/placing, or re-orienting objects before a final placement). Compared to the 10-task suites above, LIBERO-Long places greater emphasis on *temporal composition* of skills and robustness over extended roll-outs. We train and evaluate our policies on the **LIBERO-10** suites, following the standard 10-task protocol in the benchmark [15].

**Demonstrations and dataset regeneration.** LIBERO provides a small set of high-quality human demonstrations for each task. In the benchmark's reference setup, each task is paired with 50 teleoperated trajectories collected by human experts. In our experiments, following [4], we regenerate (replay) the released demonstration trajectories in simulation and re-render the corresponding observations under our camera configuration. We further filter the regenerated dataset by removing failed demonstrations that do not satisfy the task success condition. We use overall 1711 filtered demonstration trajectories for training.

**Policies and Baselines.** For single-task experiments, we adopt Diffusion Policy (**DP**) [1] as our base visuomotor policy. For multi-task experiments, we use **SmolVLA** [8] to train a Vision-Language-Action model. Notably, SmolVLA is typically trained by freezing the pretrained vision-language backbone and optimizing only the action expert. In contrast, AxisGuide requires adapting the visual representation to effectively apply the additional coordinate cue channels. For fair comparison, we train the full model including the image backbone. This is a standard and often necessary choice when training VLAs, where adapting the visual encoder is crucial for grounding action prediction in task- and domain-specific visual features, as done in [4], [5], [23]. All models are trained with identical inputs and pipelines, differing only in the use of coordinate cue images.

*C. Does AxisGuide Help the Policy Better Exploit Multi-View Observations?*

We evaluate **AxisGuide** in a multi-view setting with a fixed front camera and a wrist-mounted camera. While the front view provides a relatively consistent correspondence between image-space directions and the robot base-frame axes, the wrist view continuously changes this correspondence as the end-effector moves. As a result, a policy must interpret wrist-view observations in a coordinate-consistent way to act correctly. In this section, we show that AxisGuide remains reliable in this dynamically changing setting by providing an explicit axis reference in each view, enabling robust use of wrist-view observations. We first study a single-task setting to show the benefits of coordinate cues in multi-view control, and then extend to a multi-task setting to test whether the gains persist as task diversity increases. All training and evaluation follow the same setup as Sec. III-B.

**Single-Task Setting.** In simulation, we select (1) **Pick & Place (Bowl)**, (2) **Put Both**: putting both the cream cheese box and the butter in the basket (LIBERO-Long), and (3) **Stove (Long)**: turning on the stove and putting the moka pot on it (LIBERO-Long). We choose harder tasks than those in the simulation setting of Sec. III-B, since adding a wrist-mounted camera already improves success rates without modifying the training pipeline. For real-world evaluation, we follow the real-world task setup in Sec. III-B.

As shown in Fig. 7, AxisGuide improves success rate on **Pick & Place (Bowl)** from 74.7% to 88.0% (+13.3%p) and on **Put Both** from 78.8% to 86.7% (+7.9%p) in simulation. In real-world tasks, As summarized in Table II (Multi-view), AxisGuide improves performance on all evaluated real-world tasks, boosting success on **Pick & Place (Grape)** by 3.27%p, **Flip Pot** by 20.00%p, and **Close Pot** by 16.67%p. Overall, we find that AxisGuide yields larger gains on tasks that benefit most from wrist-view, such as **Flip Pot** and **Close**

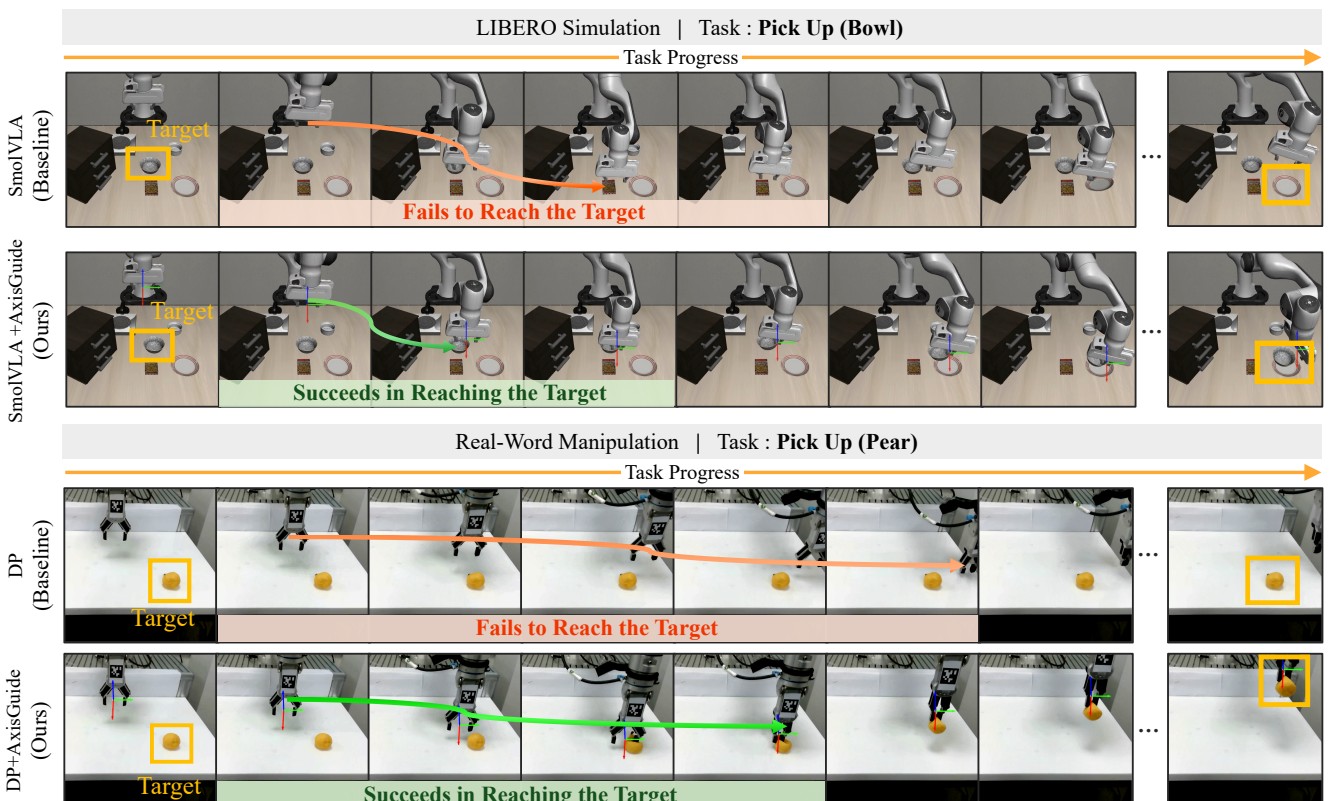

Fig. 8: **Rollout Behaviors Under Unseen Object Locations on LIBERO Simulation and Real-World Manipulation.** In the **Pick Up (Bowl)** task in the LIBERO simulation (top), the baseline model (SmolVLA) fails to adapt its actions when the target bowl is placed at an unseen location. In contrast, the same model trained with AxisGuide precisely reaches the target by grounding the action coordinate system in the image-space. A similar trend is observed in the real-world **Pick Up (Pear)** task (bottom), where DP serves as the baseline.

**Pot**.

**Multi-Task Setting.** As task diversity scales, a single language-conditioned policy must execute many instruction-following behaviors across diverse object configurations, making robust action-coordinate grounding increasingly important. We therefore evaluate whether AxisGuide remains effective in this multi-task setting. We train a SmolVLA [8] model on 10 tasks from each LIBERO suite (LIBERO-Spatial, LIBERO-Goal, LIBERO-Object, and LIBERO-Long). For a fair comparison, we include two SmolVLA baselines trained with the same pipeline: (i) action-expert-only with a frozen VLM backbone, and (ii) full fine-tuning. We run 20 rollouts per task and report the suite-level average success rate aggregated over all 200 rollouts (10 tasks × 20 rollouts) in each suite. We observe consistent gains in the multi-task setting across all four LIBERO suites, as shown on the right of Fig 7. Notably, SmolVLA trained with AxisGuide outperforms the fully trained SmolVLA by 7.5% on LIBERO-Object. These results suggest that AxisGuide provides useful action coordinate grounding even when the policy must map diverse language instructions to low-level actions across many tasks. In other words, the gains in the multi-task setting show that AxisGuide scales beyond single-task behavior cloning: it helps the policy maintain a consistent interpretation of action dimensions under increased task diversity, leading to more robust instruction-conditioned execution even in multi-view settings.

### D. Ablation Study

We ablate three key design choices in AxisGuide: *(i) how* the coordinate cues are injected into the visual input, *(ii) where* the cues are positioned in the image, and *(iii) whether* the projected action directions are normalized. Specifically, we compare (a) overlaying the axes on top of the RGB image versus concatenating the cue as additional input channels, (b) placing the cue in a fixed, center-aligned location versus EEF-aligned positioning, where the cue is anchored to the end-effector location in the image to reflect the action-centric viewpoint, and (c) using raw projected vectors versus normalized unit directions. We evaluate all variants on the **Pick & Place (Bowl)** task using the same Diffusion Policy backbone and training protocol in simulation, and report success rates.

As shown in Table III, adding AxisGuide cues improves performance across all variants, confirming that explicit coordinate grounding is beneficial. However, the magnitude of improvement depends strongly on the injection and positioning choices. Overlay + EEF-aligned (v1) improves success from 74.67% to 84.00% (+9.33%p), suggesting that even a pure visual overlay can provide useful directional

TABLE III: **Ablation of AxisGuide design choices.** Average success rates (%) over 75 rollouts on **Pick & Place (Bowl)**. **Inject.** denotes how the visual cue is incorporated into the input (overlay on RGB vs. channel-wise concatenation). **Pos.** indicates how the cue is positioned in the image (eef: aligned with the end-effector; center: image-centered). **Norm.** specifies whether the projected action directions are normalized to unit vectors.

| Variant | Inject. | Pos. | Norm. | Success (%) |
|---|---|---|---|---|
| DP (Baseline) | – | – | – | 74.67 |
| DP + AxisGuide (v1) | overlay | eef | yes | 84.00 (+9.33) |
| DP + AxisGuide (v2) | concat | center | yes | 80.00 (+5.33) |
| DP + AxisGuide (v3) | concat | eef | no | 85.30 (+10.63) |
| **DP + AxisGuide (Ours)** | **concat** | **eef** | **yes** | **88.00 (+13.33)** |

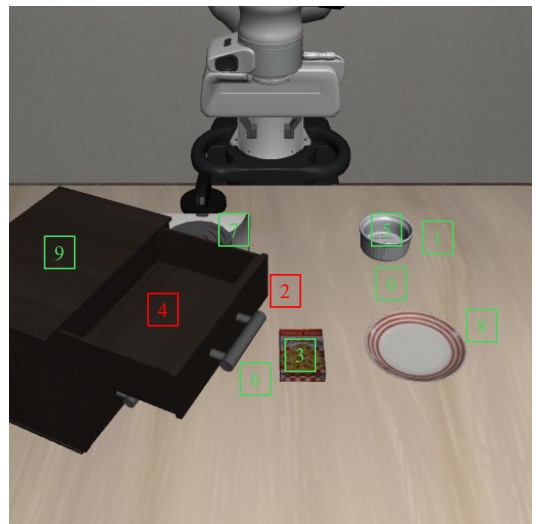

Fig. 9: **LIBERO-Spatial task setup.** Numbered boxes indicate the typical target-object region for each of the ten LIBERO-Spatial tasks (0–9), where demonstrations place the object near the corresponding region. For our *object novel position* study, we train on the remaining regions (green) while excluding tasks 2 and 4 (red). At evaluation time, we progressively expand the test placement region outward from the task-2 region to cover the full tabletop workspace, measuring generalization to unseen object locations.

information when it is localized around the region where actions are executed. In contrast, Concat + Center-aligned (v2) achieves 80.00% (+5.33%p), indicating that simply providing coordinate cues is not sufficient. Their spatial placement matters, and placing the cue at the image center can be less informative when the relevant interaction occurs away from the center. Finally, we observe that normalizing the projected action directions further improves performance. Without normalization (v3), the success rate drops to 85.30%, compared to 88.00% with normalization. This suggests that removing scale variations caused by depth and camera geometry allows the policy to focus on directional information, leading to more stable and consistent behavior.

Our final design, Concat + EEF-aligned with normalization, yields the best performance, reaching 88.00% success

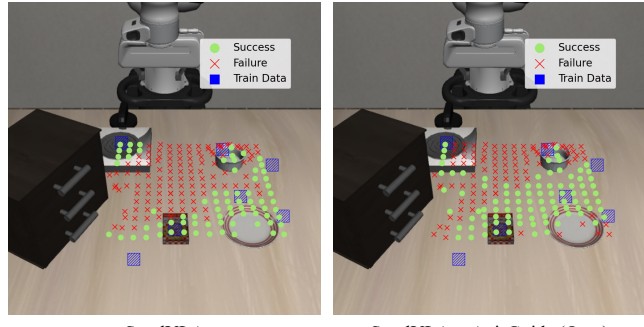

SmolVLA  SmolVLA + AxisGuide (Ours)

(a) Spatial Generalization Across Object Positions in Simulation

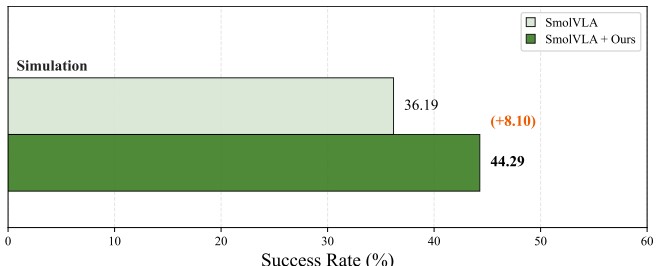

(b) Comparison of Success Rates in Simulation (Front Cam Only)

Fig. 10: **Generalization to Novel Object Positions. (Front Cam Only)** (a) shows that the baseline SmolVLA (left) generalizes poorly, with success largely confined to regions near training data, whereas SmolVLA with AxisGuide (right) reliably reaches unseen object positions between clusters, which is consistent with the multi-view results in Fig. 3. We report corresponding success rates in (b), where AxisGuide delivers substantial gains in simulation settings.

rate (+13.33%p over the baseline). This shows that how we inject the cue matters: compared to a pure overlay, channel concatenation provides the model with additional cue channels without altering or occluding the original RGB content, enabling the backbone to access both the raw appearance information and the coordinate cues in a clean, disentangled representation. Moreover, anchoring the cue at the end-effector further improves performance by placing the coordinate reference exactly where actions are executed, making the cue most relevant to the local interaction. We additionally find that normalizing the projected action directions further improves performance by removing scale variations, allowing the policy to focus on directional action semantics. Overall, these results motivate our final design choice of injecting AxisGuide cues via concatenated channels with EEF-aligned positioning and normalized directions.

### E. Object Novel Position Experiment Details and Additional Single-View Evaluation

**Experiment Details.** We begin from the LIBERO-Spatial [15] suite, which is designed to test spatial knowledge transfer across tasks. LIBERO is widely used in the manipulation community and provides high-quality human demonstrations for each task, making it a convenient and reproducible starting point for constructing controlled gen-

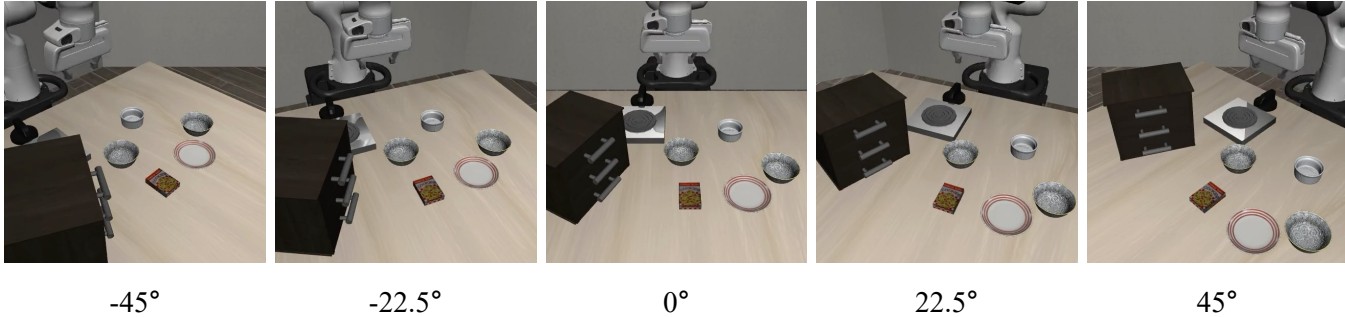

| -45° | -22.5° | 0° | 22.5° | 45° |
|------|--------|-----|-------|-----|

Fig. 11: **Viewpoint generalizability task setup.** Visualization of the dataset constructed by varying the camera viewpoint from $-45°$ to $45°$ in $22.5°$ increments. We use demonstrations from LIBERO-Spatial tasks 0 and 2 to train SmolVLA [8], and evaluate generalization to unseen viewpoints by testing viewpoints at $10°$ intervals.

eralization tests. However, in the standard LIBERO-Spatial setting, the target object location within each task does not vary substantially, making it difficult to directly evaluate whether a policy can *adapt its behavior* when the same object is moved to a previously unseen position. To explicitly test robustness to object relocation, we construct a modified benchmark based on LIBERO-Spatial. In LIBERO-Spatial, the target object is the *black bowl*. Since the scene contains two identical black bowls, we remove one bowl from the scene to eliminate ambiguity about which instance should be manipulated. To illustrate the spatial variation across tasks, we visualize the target black-bowl location for each LIBERO-Spatial task in Fig. 9. Based on this visualization, we exclude Task 2 and Task 4 from training (corresponding to the cases where the black bowl is placed at the table center or inside the drawer), and train policies only on the remaining 8 tasks. This modification increases the diversity of the black-bowl placement across tasks, and also introduces mild intra-task variation in the bowl position via randomized initial states. For evaluation, we perform rollouts where the episode starts with the black bowl placed at the table center, and we sweep the initial bowl position across the entire tabletop workspace.

This setting forms a new benchmark that directly tests whether the policy can visually localize the object under relocation and produce actions that correctly track and approach it under unseen spatial configurations. To measure whether the policy successfully follows the relocated object, we define success as establishing contact between the robot gripper and the black bowl (i.e., gripper-bowl contact is treated as the success condition).

**Single-view Evaluation (front camera only).** In the main paper, we report object relocation performance in a multi-view setting. To verify that AxisGuide also improves action understanding and object relocation robustness under a single external view, we conduct an additional experiment where the policy is trained and evaluated using only the fixed front camera. As shown in Fig 10, AxisGuide improves the success rate from 36.19% to 44.29% (+8.10%p), indicating that the benefits of AxisGuide carry over to the front-camera-only setting.

TABLE IV: **Quantitative Comparison of Viewpoint Generalizability in the LIBERO Simulation.** We evaluate average success rates (%) over 20 rollouts.

| Angle | SmolVLA [8] | SmolVLA + KYC [17] | SmolVLA + Ours |
|-------|-------------|--------------------|-----------------|
| -40° | 65.00 | 67.50 | **72.50** |
| -30° | 65.00 | **82.50** | 67.50 |
| -20° | 77.50 | 65.00 | **85.00** |
| -10° | **82.50** | **82.50** | **82.50** |
| 0° | 72.50 | **77.50** | **77.50** |
| 10° | 70.00 | **82.50** | 75.00 |
| 20° | 82.50 | 87.50 | **90.00** |
| 30° | 60.00 | 67.50 | **77.50** |
| 40° | 72.50 | 82.50 | **95.00** |
| Avg. | 71.94 | 77.22 | **80.28** |

### F. Viewpoint Generalizability in Single-View Training

**Experiment Details.** Many large-scale robot datasets contain demonstrations collected under diverse camera viewpoints, and a practical policy should remain robust when the same task is observed from novel angles. While our main paper evaluates (i) a *single-view* setup where the action coordinate semantics are fixed with respect to a static external camera, and (ii) a *multi-view* setup where a wrist-mounted camera introduces dynamically changing viewpoints, here we explicitly test whether AxisGuide remains effective when training data spans *multiple camera angles* even under a single-view policy. To this end, we construct a controlled viewpoint generalizability benchmark in LIBERO.

**Experiment.** We use LIBERO task instances (task 0 and 2) and collect training demonstrations while rotating the front camera by $22.5°$ increments from $-45°$ to $+45°$. We then train policies on this multi-viewpoint dataset and evaluate them on unseen viewpoints by testing at $10°$ intervals for tasks 0 and 2. We visualize the collected viewpoints and task setup in Fig. 11. To directly assess viewpoint invariance, we use *only the front camera* as the policy input in both training and evaluation.

**Results.** As shown in Table IV, AxisGuide achieves the best success rates on most evaluation angles (all except $-30°$

TABLE V: **Runtime and model size overhead of Axis-Guide on SmolVLA.** End-to-end latency (batch=1) includes AxisGuide cue generation and input concatenation. Axis-Guide adds only **+5.42 ms** ($\approx 0.005$ s) overhead.

| Method | Total params (M) $\downarrow$ | $\Delta$Latency (ms) $\downarrow$ |
|---|---|---|
| SmolVLA (Vanilla) | 450.046 | 0.00 |
| SmolVLA + AxisGuide (Ours) | 450.636 | +5.41 |
| Overhead (Ours $-$ Vanilla) | **+0.590 (+0.13%)** | **+5.41** |

TABLE VI: **Calibration sensitivity analysis.** We evaluate robustness to calibration errors by injecting extrinsic perturbations (camera translation and rotation, left) and intrinsic perturbations (focal length and principal point offsets, right) at inference time, while training with unperturbed parameters.

| Rot. \ Trans. | 0 cm | 1 cm | 2 cm | 3 cm | Focal. \ P.P. | 0 px | 5 px | 10 px |
|---|---|---|---|---|---|---|---|---|
| 0° | **65.7** | 64.3 | 61.9 | 62.4 | 0% | **65.7** | 63.8 | 65.2 |
| 3° | 64.8 | 64.8 | 62.9 | 61.4 | 5% | 64.3 | 63.8 | 65.2 |
| 6° | 64.3 | 64.8 | 64.3 | 63.3 | 10% | 64.3 | 64.3 | 63.8 |

and $+10°$). On average across viewpoints, AxisGuide improves performance by **+8.34**%p over vanilla SmolVLA and by **+3.04**%p over the viewpoint-robust baseline KYC [17]. These results indicate that AxisGuide remains effective when demonstrations are collected from diverse viewpoints and improves robustness to novel camera angles at test time.

### G. Efficiency Analysis

We analyze the computational overhead introduced by AxisGuide when applied to SmolVLA. Table V reports both the additional learnable parameters and the end-to-end latency increase (batch=1), where latency includes AxisGuide cue generation and channel-wise concatenation. All measurements are conducted under the multi-view setting using both a wrist-mounted camera and a fixed front camera.

AxisGuide adds only **0.590M** parameters on top of the **450.046M** parameters of the vanilla model, corresponding to a negligible **+0.13%** increase. More importantly, the runtime overhead is minimal: AxisGuide incurs only **+5.41 ms** additional latency per inference, i.e., approximately **0.005 s**. These results suggest that AxisGuide provides explicit action-coordinate grounding with *negligible* compute and model-size overhead, making it practical to deploy as a lightweight visual cue to existing manipulation policies.

### H. Calibration Sensitivity Analysis

We further evaluate the robustness of our method to projection errors caused by inaccurate system calibrations on the Novel Object Position task in Fig. 3. In the Table VI, we report results under extrinsic (left) and intrinsic (right) perturbations at inference time, while using a model trained with unperturbed camera parameters. Errors in robot kinematics and EEF pose estimations are reflected in this experiment, since they also perturb the projected EEF position. These results demonstrate our method's **robustness to realistic calibration noise**, as performance degrades marginally and still exceeds the baseline [23] performance of 52.4 reported under unperturbed settings.

### I. Comparison with Visual Cue Baselines

We compare our method with prior image-cue baselines [24], [25] on the Novel Object Position task in Fig. 3. TraceVLA [25] and AimBot [24] aim to improve scene understanding by providing visual cues that encode the spatial relationship between the object and the end-effector. However, successful task execution in novel environments

requires more than understanding the spatial relationship. The policy must also determine what action to take in the novel scene, which requires understanding the robot's action coordinate system in image space. While TraceVLA and AimBot enrich the observation with scene-level spatial cues, they are not specifically designed to explicitly represent the robot's action coordinate system. As a result, when the object appears in unseen locations, these methods offer limited guidance on how the policy should adjust its actions, making improvements under distribution shift difficult. In contrast, by explicitly grounding the robot's action coordinates in the visual observation and showing where the $x$, $y$, and $z$ action directions lie in the image, AxisGuide helps the policy infer how to move toward a target even under novel object positions. This leads to larger gains in Table VII and enables more reliable, robust manipulation.

TABLE VII: **Comparison with visual cue baselines on the Novel Object Position task.** We report success rate (%) for SmolVLA and its variants augmented with different visual cues.

| Method | Success Rate (%) |
|---|---|
| SmolVLA [8] | 52.38 |
| + TraceVLA [25] | 51.90 |
| + AimBot [24] | 52.38 |
| + AxisGuide (Ours) | **65.71** |

## VII. RELATED WORK

### A. Visuomotor Policies for Manipulation

Recent work has substantially advanced visuomotor policies for manipulation that map RGB observations directly to actions. Diffusion Policy [1] formulates action generation as a stochastic denoising process over trajectories to capture multimodal action distributions. Vision-Language-Action (VLA) models [10], [6], [4] improve generalization to novel objects and semantically varied instructions by conditioning on language.

One way to represent robot actions is to use the joint space, where actions are defined over joint configurations and a low-level controller supplies the required dynamics. However, such actions are difficult to align with image observations because individual joints are not explicitly visible. An alternative, adopted by most open-source robotic manipulation datasets [11], [12], is to use relative end-effector displacements defined in the Cartesian space of the

robot base frame. Following prevalent practice [10], [6], [4], we train policy models to predict relative Cartesian displacements of the end effector.

Under this relative coordinate system, we investigate whether providing a direct visual reference for the action coordinate system in image space alleviates the difficulty of learning to identify the robot's base frame from images.

### B. Additional Inputs for Visuomotor Policies

Visuomotor policies are typically trained to predict actions from RGB observations and a task specification, yet this input alone frequently fails to yield robust generalization. Recent work therefore introduces additional inputs that act as visual intermediaries or auxiliary context. RT-Trajectory [16] replaces ambiguous language with coarse 2D trajectory sketches to better specify intent and improve generalization. RoboPoint [26] uses a VLM to predict keypoint-like affordances on the image and leverages them to guide low-level control. Several methods also directly augment pixels with extra spatial or temporal cues. To compensate for the lack of temporal and spatial information in image observations, TraceVLA [25] overlays tracked motion traces to encode recent history. AimBot [24] renders reticles and lines using depth and camera geometry providing explicit spatial cues about the object-EEF relationship, and HAMSTER [27] overlays intermediate 2D paths predicted by VLMs to separate high-level planning from motor execution. Complementary to these, other approaches [28], [17] condition policies on hardware context by soft-prompting embodiment-specific configurations or explicitly providing camera geometry to improve robustness across embodiments and sensor setups.

While these techniques enrich observations with task intent, spatiotemporal context, or hardware information, they still do not explicitly consider how the policy's action coordinate system (e.g., "+x translation" or "+z rotation" in the robot frame) should be interpreted in the given image space. AxisGuide addresses this missing component by augmenting RGB observations with an explicit visualization of the action coordinate system in the 2D image space, thereby grounding action semantics in pixel space. Different from prior input-augmentation strategies, Our work explicitly injects action coordinate system information as an image-space cue and represents a new direction for improving the generalization of visuomotor policies.

## VIII. Limitations

First, AxisGuide assumes access to camera parameters which are used to project the base-frame axes into each view. While such information is available in many simulation benchmarks and can be measured or calibrated in real-world systems, it may not be readily provided in all datasets or deployment environments. Fortunately, large-scale datasets such as DROID [12] include calibrated multi-view setups, suggesting that this requirement is not prohibitive. Moreover, when calibration metadata is missing, recent camera-to-robot pose estimation methods [29], [30] could be used to estimate the required extrinsics, potentially addressing this limitation.

Second, we observe that even when AxisGuide successfully tracks objects at novel locations, failures can still occur in contact-rich stages such as grasp execution, suggesting that coordinate grounding does not resolve all sources of error. Addressing such failures may require complementary improvements in perception of contacts, gripper-object interactions, or reward signals.

Finally, our evaluation, while spanning multiple tasks and including real-world experiments, is still limited in scale compared to massive robot datasets. A more scalable study on large, diverse datasets (e.g., DROID [12]) would further clarify how AxisGuide behaves under broader distribution shifts and whether the gains persist when training is scaled up.

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
