# OpenReview forum: "AxisGuide: Grounding Robot Action Coordinate System in RGB Observations for Robust Visuomotor Manipulation"
_IEEE.org/ICRA/2026/Workshop/Manipulation_Robustness — ICRA 2026_

### Official Review · Reviewer_vaiQ · 2026-05-03
**A simple and effective method for improving visuomotor robustness**

**Rating:** 8
**Confidence:** 4

**Review:**

This paper proposes AxisGuide, a lightweight method that renders robot action axes as additional cues for visuomotor manipulation policies. The core claim is that explicit action-coordinate grounding improves robustness to object-location and viewpoint shifts. The authors report consistent gains in LIBERO simulation and real-world experiments with additional ablations on key design choices.

Strengths

1. The problem is timely and well motivated, especially for robustness visuomotor manipulation.
2. The proposed method is simple, interpretable, and easy to integrate into existing image-to-action policies.
3. The experiments are carefully designed and provide useful evidence for the core claim. The ablation study is also informative and supports the main design choices.

Weaknesses

1. The method relies on accurate camera calibration and end-effector poses. The paper would benefit from further analysis of its sensitivity to calibration noise or pose-estimation errors.
2. Some tasks have relatively high baseline success rates, making the performance gains somehow marginal to me.

Overall,  the authors present an interesting idea with convincing empirical evidence.

---

### Decision · Program_Chairs · 2026-05-21

Accept